# Thermal Decomposition Mechanism and Kinetics Study of Plastic Waste Chlorinated Polyvinyl Chloride

**DOI:** 10.3390/polym11122080

**Published:** 2019-12-12

**Authors:** Ru Zhou, Biqing Huang, Yanming Ding, Wenjuan Li, Jingjing Mu

**Affiliations:** 1Jiangsu Key Laboratory of Urban and Industrial Safety, College of Safety Science and Engineering, Nanjing Tech University, Nanjing 211816, China; 201761100038@njtech.edu.cn (W.L.); 201861201043@njtech.edu.cn (J.M.); 2Faculty of Engineering, China University of Geosciences, Wuhan 430074, China; 1201820343@cug.edu.cn (B.H.); dingym@cug.edu.cn (Y.D.)

**Keywords:** chlorinated polyvinyl chloride, pyrolysis, thermogravimetry, plastic waste, reaction mechanism, kinetics

## Abstract

Chlorinated polyvinyl chloride (CPVC), as a new type of engineering plastic waste, has been used widely due to its good heat resistance, mechanical properties and corrosion resistance, while it has become an important part of solid waste. The pyrolysis behaviors of CPVC waste were analyzed based on thermogravimetric experiments to explore its reaction mechanism. Compared with polyvinyl chloride (PVC) pyrolysis, CPVC pyrolysis mechanism was divided into two stages and speculated to be dominated by the dehydrochlorination and cyclization/aromatization processes. A common model-free method, Flynn-Wall-Ozawa method, was applied to estimate the activation energy values at different conversion rates. Meanwhile, a typical model-fitting method, Coats-Redfern method, was used to predict the possible reaction model by the comparison of activation energy obtained from model-free method, thereby the first order reaction-order model and fourth order reaction-order model were established corresponding to these two stages. Eventually, based on the initial kinetic parameter values computed by model-free method and reaction model established by model-fitting method, kinetic parameters were optimized by Shuffled Complex Evolution algorithm and further applied to predict the CPVC pyrolysis behaviors during the whole temperature range.

## 1. Introduction

Global plastics production has increased over the years due to the widespread use in many fields. It is very convenient for people to use plastic products due to their resistance to degradation, versatility, light weight and low cost [1]. Chlorinated polyvinyl chloride (CPVC), a new type of engineering plastics with broad application prospects, is obtained by further chlorination of polyvinyl chloride (PVC). When the PVC is chlorinated, its physical and chemical performance, such as irregularity, polarity, solubility and chemical stability of molecular chain arrangement, can be improved [2]. Especially, due to its good heat resistance, mechanical properties and corrosion resistance, CPVC is widely used in various industrial fields. However, the short cycle life of plastics leads to the emergence of a large number of plastic wastes [3]. How to solve the waste CPVC in a scientific, reasonable and effective way? There is no doubt that it is an important issue that needs to be solved urgently. For the treatment of plastic waste, converting it from waste to new energy is a big direction in future waste utilization [4]. Pyrolysis, as a very promising thermochemical technique [5], has been paid more and more attention by providing an excellent alternative to transform plastic wastes into energy fuels or valuable chemicals [6,7], especially decomposing long-chain polymer molecules into smaller, less complex molecules [8]. It not only overcomes the drawbacks of landfill and incineration but also recovers valuable fuel or chemical raw materials from solid waste [9].

Some typical plastic wastes in the industry have been researched for their thermal degradation process. Miranda et al. [10,11] studied the pyrolysis process of PVC in vacuum environment and made the kinetic and product analysis. Gui et al. [12] conducted a pyrolysis experiment of PVC to study the effects of peak temperature, holding time and heating rate on the formation of nascent tar. Al-Salem and Liu et al. [13,14] studied pyrolysis kinetics of the high density polyethylene. Luyt [15] investigated the effect of six halogen-free flame retardant (FR) formulations on the thermal stability of two low-density polyethylenes (LDPE) and one linear low-density polyethylene (LLDPE). Park et al. [16] characterized the pyrolysis of waste polyethylene using two successive stages with auger and fluidized bed reactors. Swann et al. [17] revealed the pyrolysis and combustion of rigid PVC using a two-dimensional model. Sun et al. [18] systematically studied the thermal decomposition characteristics of PVC using thermogravimetry coupled with mass spectrometer, while Wu et al. [19] analyzed the co-pyrolysis behavior of polyethylene, polystyrene and PVC under nitrogen atmosphere by thermogravimetry coupled with Fourier transform infrared spectroscopy. Wang [20] investigated the influence of methacryl-functionalized polyhedral oligomeric silsesquioxane (MA-POSS) nanoparticles as a plasticizer and thermal stabilizer for a PVC homopolymer and for a PVC/dissononyl cyclohexane-1,2-dicarboxylate binary blend system.

Although it is important of engineering plastic waste pyrolysis for energy utilization, there are few studies on the CPVC pyrolysis. Elakesh et al. [21,22] performed a thermogravimetric analysis of CPVC at various heating rates in the nitrogen, air and oxygen atmosphere compared with PVC. Carty et al. [23] compared the thermal decomposition of one unplasticized CPVC and three plasticized CPVC. However, the above researches are not enough to reveal the pyrolysis dynamics of engineering plastic waste CPVC, so the aim of current paper is to explore its pyrolysis behaviors and obtained appropriate kinetic parameters.

Knowledge of pyrolysis kinetics can help provide better understanding and planning of important industrial processes [24], such as their application in pyrolysis model [25] and direct combustion [26]. Model-fitting and model-fitting methods are the common ways to explore the kinetic parameters. Model-fitting method consists of fitting different models to the experimental data for the best statistical fit but with the inability to determine the reaction model [27]. However, model-free method can come over this problem without prior knowledge of the reaction model and estimate the kinetic parameters at specific extent of conversion to provide appropriate search ranges for model-fitting method [28]. Then it is recommended to explore the kinetic parameters by coupling both the model-fitting and model-free method.

Then in our current paper, the CPVC degradation process was conducted by thermogravimetric experiment and analyzed by representative model-free and model-fitting methods to explore its possible reaction model. Meanwhile, by coupling the model-free and model-fitting methods, a global optimum algorithm called Shuffled Complex Evolution (SCE) is applied to optimize kinetic parameters based on the initial kinetic parameter values computed by model-free method and reaction model established by model-fitting method. Eventually, the optimized kinetic parameters can be used to predict the pyrolysis behaviors during the whole temperature range.

## 2. Material and Methods

### 2.1. Elemental Analysis

The elemental analysis of CPVC was conducted by the element Vario EL cube (Elementar, Langenselbold, Germany) and the percentage of each element was as follows—C (31.98%), H (4.11%), N (0.06%) and Cl (63.85%, obtained by difference).

### 2.2. Thermogravimetric Eexperiments

A SDT Q600 thermal analyzer (TA Instruments, New Castle, DE, USA) was used to perform thermogravimetric experiments from 400 K to 900 K at four different heating rates (10, 20, 30 and 60 K/min). For all experimental runs, a powder sample with about 6 mg was evenly distributed in an alumina cup without a lid and a purge stream of 100 mL/min pure nitrogen was applied to the system throughout the process.

### 2.3. Kinetic Theory

Basic equation used for kinetic analysis of CPVC can be assumed based on conversion rate as follows:(1)$$\frac{d\alpha }{dt}=k(T)f(\alpha )$$(2)$$\alpha =\frac{m_{0}-m_{t}}{m_{0}-m_{f}}$$(3)$$k(T)=A\exp\left(\frac{-E_{a}}{RT}\right),$$
where *α* is the conversion rate during pyrolysis, *k(T)* is the reaction rate constant and *f(α)* is the function of reaction mechanism. *m*_0_, *m_t_* and *m_f_* represent the sample mass at the initial time, intermediate time and the end, respectively. *A* is the pre-exponential factor and *E_a_* is the activation energy of the reaction. R is the universal gas constant and *T* is the reaction temperature.

For non-isothermal pyrolysis, the constant heating rate *β* is equal to *dT*/*dt*, so Equation (1) can be written as:(4)$$\frac{d\alpha }{dT}=\frac{A}{\beta }f(\alpha )\exp(\frac{-E_{a}}{RT})$$

Equation (4) can also be expressed in integral form:(5)$$g(\alpha )=\int_{0}^{\alpha }\frac{d\alpha }{f(\alpha )}=\frac{A}{\beta }\int_{T_{0}}^{T}\exp^{-(\frac{E_{a}}{RT})}dT$$
where *g(α)* is an integral form of reaction model, *T*_0_ is the initial temperature. 

The expressions of *f(α)*, *g(α)* based on five major types of reaction models (power law models, nucleation models, reaction-order models, diffusion models and geometrical contraction models) are shown in Table 1 [29,30,31].

Both the model-free method and the model-fitting method are used in this paper, which are represented by Flynn-Wall-Ozawa (FWO) [32,33] and Coats-Redfern (CR) [34] methods. The FWO method requires the measurement of the temperatures corresponding to fixed conversion rates from experiments at different heating rates and then obtains the activation energy of a solid-state reaction without knowing the reaction model in advance. The CR method is used to calculate activation energy based on hypothetical reaction models. In other words, CR is used to calculate the activation energy from the proposed g(α) forms. Then the activation energy estimated by CR is compared to the previously obtained value by FWO to estimate the most appropriate reaction model g(α). Furthermore, based on the initial kinetic parameter values computed by FWO and reaction model established by CR, a global optimum algorithm called Shuffled Complex Evolution (SCE) is coupled to optimize the kinetic parameters.

#### 2.3.1. Flynn-Wall-Ozawa Method

The FWO method is derived by Doyle’s approximation [34] and the reaction rate in logarithmic form can be expressed as:(6)$$\ln\beta =\ln\left(\frac{AE_{a}}{Rg\left(\alpha \right)}\right)-5.331-1.052\left(\frac{E_{a}}{RT}\right).$$

The plot of ln*β* versus 1/*T* gives a straight line whose slope can be used to determine the activation energy *E_a_*. If *g(α)* was informed, the pre-exponential factor *A* can be gained.

#### 2.3.2. Coats-Redfern Method

The basic equation for CR method is given below:(7)$$\ln\left[\frac{g(\alpha )}{T^{2}}\right]=\ln\frac{AR}{\beta E_{a}}(1-\frac{2RT}{E_{a}})-\frac{E_{a}}{RT}.$$

The activation energy *E_a_* can be obtained by the plot of ln(*g(α)*/*T*^2^) versus 1/*T* and the pre-exponential factor *A* can be obtained from the intercept of this graph. *g(α)* can be varied according to different reaction models.

By comparison between the general reaction temperature range and activation energy *E_a_*, it is found *E_a_*/R*T*>>1, (1 − 2R*T*/*E_a_*) ≈ 1 [34] and then Equation (7) can be simplified as:(8)$$\ln\frac{g\left(\alpha \right)}{T^{2}}=\ln\left(\frac{AR}{\beta E_{a}}\right)-\frac{E_{a}}{RT}.$$

Furthermore, a lot of reaction models *g(α)* are assumed and tried to compute the corresponding activation energies. If the computed activation energy by a certain reaction model is the closest to the previously obtained value by FWO, then this given reaction model is defined as the most appropriate reaction model *g(α)*.

Moreover, the kinetics compensation effect can be used to test the correctness of selected reaction model. Generally speaking, if the selected reaction model *g(α)* is suitable for characterizing solid pyrolysis, then there is a linear relationship between the natural logarithm of the pre-exponential factor ln*A* and the activation energy *E_a_* [35], as expressed in Equation (9):(9)$$\ln A=aE_{a}+b$$
where *a* and *b* are called compensation parameters.

#### 2.3.3. Shuffled Complex Evolution Optimization Method

SCE method is a robust, effective and efficient global optimum algorithm and its detailed description can be referred to Ref. [24,36,37]. In our current study, the objective function ϕ is defined by comparing the differences between the predicted results and experimental data of mass loss rate:(10)$$\phi =\sum_{j=1}^{N}\left[\frac{\sum_{k=1}^{\lambda }{\left(MLR_{\mathrm{pred},k}-MLR_{\exp,k}\right)}^{2}}{\sum_{k=1}^{\lambda }{\left(MLR_{\exp,k}-\frac{1}{\lambda }\sum_{p=1}^{\lambda }MLR_{\exp,p}\right)}^{2}}\right]$$(11)$$R^{2}=1-\phi ,$$
where *N* is the number of experiments and *λ* is the number of experimental data points for each experiment. Subscript *pred* and *exp* represent the predicted results and experimental data.

## 3. Results and Discussion

### 3.1. Thermogravimetric Analysis

The thermogravimetric curves of CPVC at four heating rates (10, 20, 30 and 60 K/min) under the nitrogen atmosphere are showed in Figure 1, including the mass loss (TG) *m*/*m*_0_ and mass loss rate (DTG) d(*m*/*m*_0_)/d*T*. The main thermal degradation of CPVC is in the temperature range of 500 K to 800 K and the final residue remains about 40%. The trends of mass loss curves corresponding to the four heating rates are basically the same. There are two obvious rapid descent zones in Figure 1a which are in the temperature range of 540–620 K (Stage I) and 710–780 K (Stage II, marked in grey dashed boxes), respectively. Wherein, the stage between them is assumed to be transition region. Just as shown in Figure 1b, two distinct peaks are closely related to the above-mentioned rapid descent zones. The temperature locations of these two peaks are about 570–610 K and 730–760 K, respectively. Furthermore, as the heating rate become faster, the value of two peaks decreases and their corresponding temperature value increases, namely slower heating rates correspond to larger mass loss rate peak values and taking place at lower peak temperatures, while they have little impact on the variation tendency of the pyrolysis.

To explore the reaction mechanism of CPVC, the pyrolysis behaviors of PVC conducted by Miranda et al. [11] is provided to compare with that of CPVC, as shown in Figure 2. The pyrolysis residue of PVC (only 10%) is far less than that of CPVC (40%) in nitrogen atmosphere. Similarly, there are two obvious pyrolysis stages of PVC and CPVC during the whole pyrolysis process. 

As shown in Figure 2, the mass loss percentage of PVC is about 55% in Stage I. Considering that the molecular structure of PVC is mainly composed of -CH_2_-CHCl- unit, the percentage of element Cl in PVC is estimated about 56.77%, which is close to the lost mass in Stage I. This phenomenon is in accordance with the experimental data of Castro et al. [38], who found that almost all the element Cl was removed from the PVC when the temperature is lower than 613 K. Compared with CPVC, the reaction rates of PVC are larger in Stage I. The lost mass percentage of CPVC is only about 40%, lower than that of PVC (55%). Especially, the peak value of PVC is nearly 1.7 times that of CPVC and its starting pyrolysis temperature is a little lower, which means that CPVC shows better chemical stability of molecular chain arrangement than PVC. However, in Stage II, the TG and DTG curves of PVC and CPVC are almost the same, with the lost mass percentage being about 20%. Namely, the pyrolysis behaviors of CPVC are almost the same with that of PVC in Stage II.

Considering CPVC is further product of PVC via chlorination, the reaction mechanism of CPVC can be speculated by that of PVC. Although CPVC is a derivative of PVC, it is a complex system. There are at least three different types of repeating units present in the polymer molecular structure: -CH_2_-CHCl-, -CHCl-CHCl- and a small amount of -CCl_2_- units [22]. Just as Liebman et al. [39] explained, the greater stability of CPVC compared to that of PVC was due to long sequences of -CHCl-CHCl- units in the polymer. Chlorination of PVC produces a change in the physical properties and thus improves the thermal properties of the polymer and makes it more stable [40], which is attributed in part to a crosslinking reaction and in part to elimination of reactive defects in the PVC structure [21], resulting in their difference in Stage I.

Huang et al. [5] emphasized that the reaction mechanism of PVC thermal degradation in nitrogen atmosphere could be attributed to two parts—in Stage I, dehydrochlorination was the main reaction of PVC decomposition, leading to the release of HCl and the formation of conjugated polyene; in Stage II, aromatic hydrocarbons were formed from cyclization reactions of the conjugated polyene and also small molecule hydrocarbons were generated [41,42,43]. Considering the similar pyrolysis behaviors of CPVC and PVC, it can be speculated that the reaction mechanism of CPVC is similar to that of PVC. Namely, in Stage I, dehydrochlorination is the main reaction leading to the release of HCl and the formation of conjugated polyene, while the more stable structure of CPVC restrains this process than that of PVC; In Stage II, aromatic hydrocarbons are formed from cyclization reactions of the conjugated polyene and also small molecule hydrocarbons are generated, namely the new formed carbon-carbon double bonds and triple bonds may be further cyclized to benzene, chlorobenzene, homologues and char and so on [44].

### 3.2. Kinetic Analysis Base on FWO Method

In order to obtain the reaction kinetics parameter activation energy, the FWO method introduced in Section 2.3.1 was employed here based on four heating rates. Meanwhile, the plots of conversion rate α versus temperature *T* are shown in Figure 3 and the linear relationships between ln*β* versus 1/*T* at various conversion rates in stages I and II are plotted in Figure 4a. Based on the Equation (6), the activation energy can be computed by the slopes. Then the trend of activation energy values corresponding to various conversion rates is shown in Figure 4b. Meanwhile, the detailed values of activation energy are listed in Table 2. It is found that the mean values of activation energy in Stage I and II are 140.27 kJ/mol and 246.07 kJ/mol, respectively. This value of Stage II is much larger than that of Stage I, which means more energy is needed in Stage II.

### 3.3. Estimation and Verification of Reaction Model

According to Equation (8), The plots of ln(*g(α)*)/*T*^2^ versus 1/*T* corresponding to various reaction mechanisms at each heating rate by CR method are obtained, with 10 K/min as example shown in Figure 5. Furthermore, the activation energies calculated by the slopes at various heating rates in Stage I and II are listed in Table 3 and Table 4.

For Stage I, the first order reaction-order model *F1*, the estimated average activation energy at four heating rates is 142.16 kJ/mol, which is the closest to the activation energy value (140.27 kJ/mol) calculated by the FWO method. So the most appropriate model should be *F1*. For Stage II, the most appropriate reaction model must be the reaction-order model, due to the value estimated by other reaction models being always less than 100 kJ/mol. Then the fourth order reaction-order model *F4* (229.76 kJ/mol) closest to the activation energy computed by FWO (246.07 kJ/mol), is chosen as the most appropriate model for Stage II.

Therefore, by the comparison of calculated value based on CR method and FWO method, the most suitable reaction models are obtained for Stage I and Stage II, namely *F1* and *F4*, respectively, corresponding to the integral forms of model functions *g(α)* = −*ln(1 − α)* and *g(α)* = *[1 − (1 − α)^(−3)^]/(−3)*. According to the intercepts fitted by the FWO method, the values of the pre-exponential factor A can be obtained by substituting *g(α)*, as listed in Table 2.

As expressed in Equation (9), Figure 6 shows the good linear relationship between ln*A* and *E_a_*, which further proves the feasibilities of the estimated reaction models. The linear relationship functions are ln*A* = 0.1822*E_a_* + 2.3442 and ln*A* = 0.1546 *E_a_* + 5.8289 for Stage I and Stage II, respectively, with *R*^2^ reaching up to 0.99.

### 3.4. Kinetic Parameters Estimation by Shuffled Complex Evolution

Based on the two established pyrolysis stages, it is assumed that the reaction can be expressed as:(12)$$\begin{array}{l} CPVC_{A}\to Residue+gas\\ CPVC_{B}\to Residue+gas \end{array}.$$

Corresponding to the first order reaction-order model and fourth order reaction-order model in the two stages established by CR method in Section 3.3, the reaction rates in these two stages can be represented:(13)$$\frac{dY}{dt}=-Y_{0}{\left(\frac{Y}{Y_{0}}\right)}^{n}A\exp\left(-\frac{E_{a}}{RT}\right)n=1\;\mathrm{in}\;\mathrm{Stage}\;I,\;n=4\;\mathrm{in}\;\mathrm{Stage}\;\mathrm{II}.$$

Next, the parameter search ranges of action energy *E_a_*, pre-exponential factor *A* and residue yield *v* should be given. Based on the calculated results by FWO method in Section 3.2, the search ranges are set to 50–150% of the calculated values. Eventually, the optimized values are obtained, as listed in Table 5. The predicted results compared to experimental data are shown in Figure 7. It can be seen that the predicted results agree well with experimental data during the whole temperature range and the *R*^2^ values reach up to 0.93 at all the heating rates. The optimized kinetic parameters can be used in the pyrolysis process of related energy utilization, such as the application of pyrolysis model and direct combustion in numerical simulation [25,26].

## 4. Conclusions

Thermogravimetric analysis experiments were carried out to study the pyrolysis of CPVC at the heating rates of 10, 20, 30 and 60 K/min in a nitrogen atmosphere. The pyrolysis process of CPVC could be regarded as two dominant stages based on the TG and DTG curves. By comparison of PVC pyrolysis, it was estimated that these two stages of CPVC pyrolysis should be mainly attributed to the dehydrochlorination and cyclization/aromatization processes, respectively.

Under the unknown CPVC pyrolysis model, a model-free method (FWO) was selected to estimate the average activation energy and then its value was compared with the activation energy values computed by a model-fitting method (CR) with different reaction models to explore the most appropriate model. Eventually, the first order reaction-order model and fourth order reaction-order model were established to be responsible for these two pyrolysis stages. Furthermore, Shuffled Complex Evolution algorithm method was coupled with FWO and CR to optimize the kinetic parameters for predicting the pyrolysis behaviors in the whole temperature range. Then these obtained kinetic parameters can be used in the further simulation about pyrolysis and combustion.

## Figures and Tables

**Figure 1 polymers-11-02080-f001:**
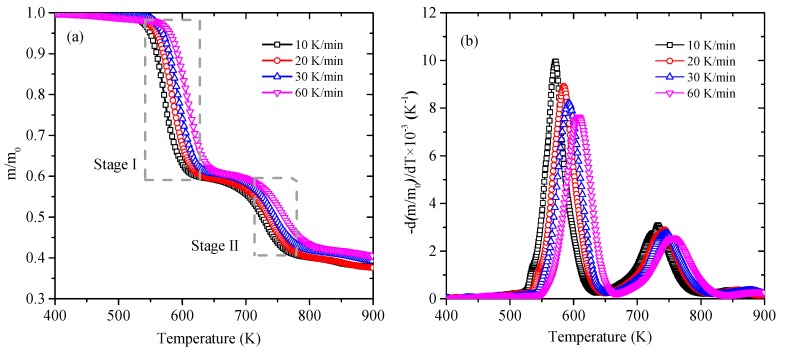
Curves of mass loss (TG) and mass loss rate (DTG) at different heating rates.

**Figure 2 polymers-11-02080-f002:**
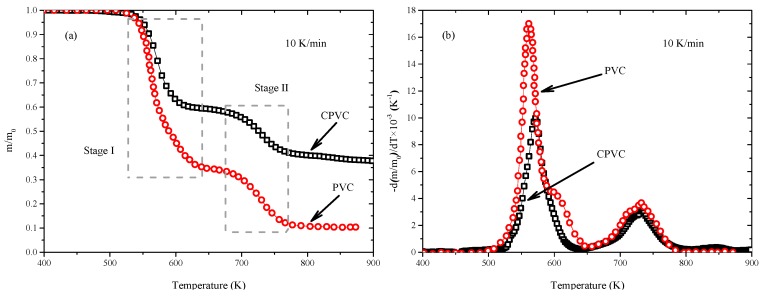
Compared TG and DTG of polyvinyl chloride (PVC) and chlorinated polyvinyl chloride (CPVC) at 10 K/min.

**Figure 3 polymers-11-02080-f003:**
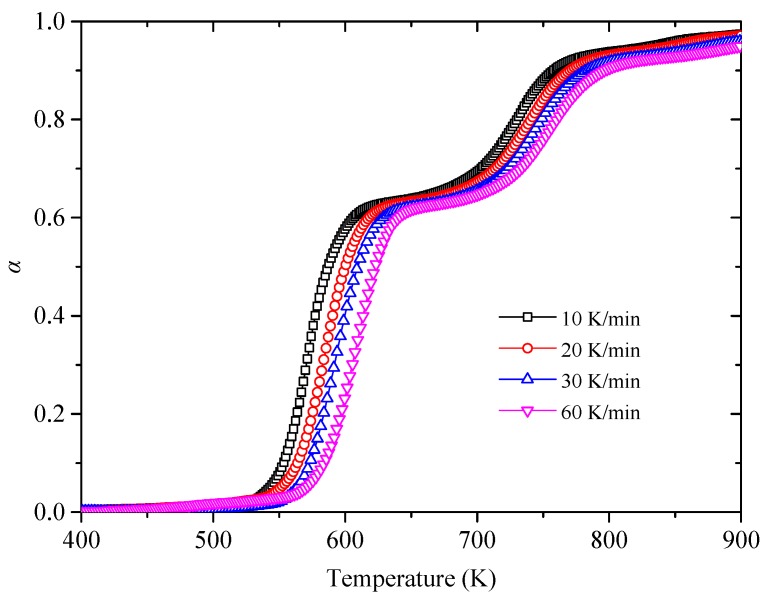
Conversion rate *α* at different heating rates.

**Figure 4 polymers-11-02080-f004:**
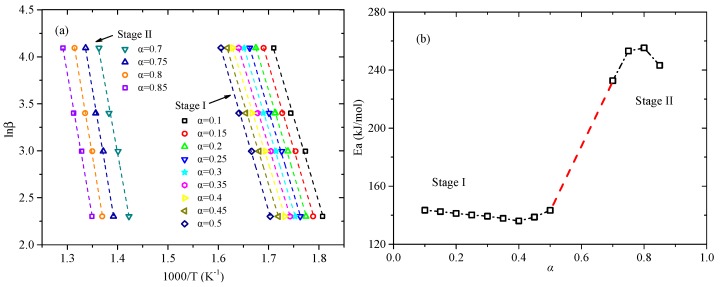
(**a**) Flynn-Wall-Ozawa (FWO) plots for different conversion rates and (**b**) activation energy trend.

**Figure 5 polymers-11-02080-f005:**
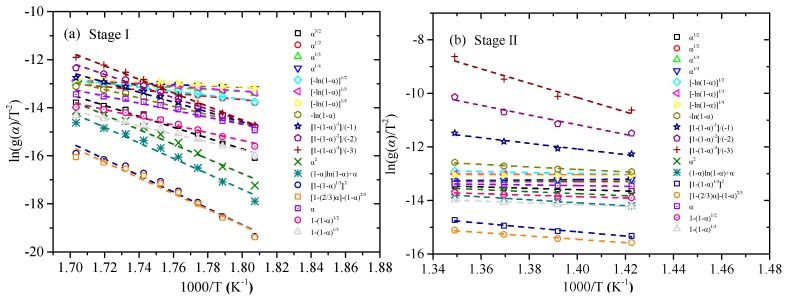
Coats-Redfern (CR) plots at 10 K/min: (**a**) Stage I and (**b**) Stage II.

**Figure 6 polymers-11-02080-f006:**
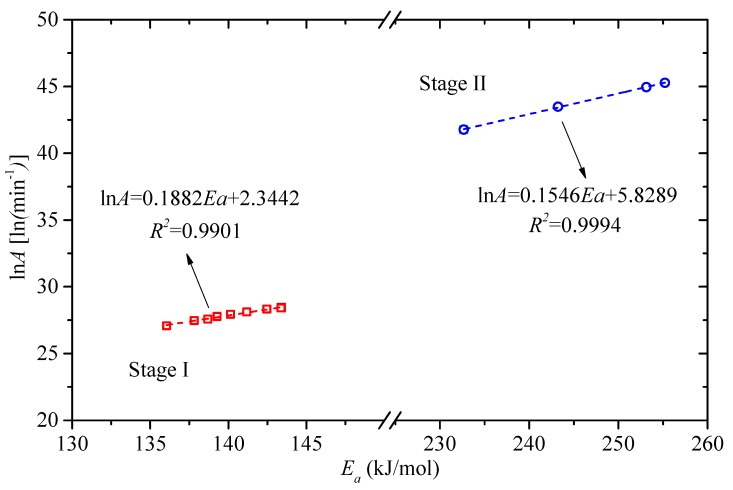
Compensation plots for FWO between ln*A* and *E_a_*.

**Figure 7 polymers-11-02080-f007:**
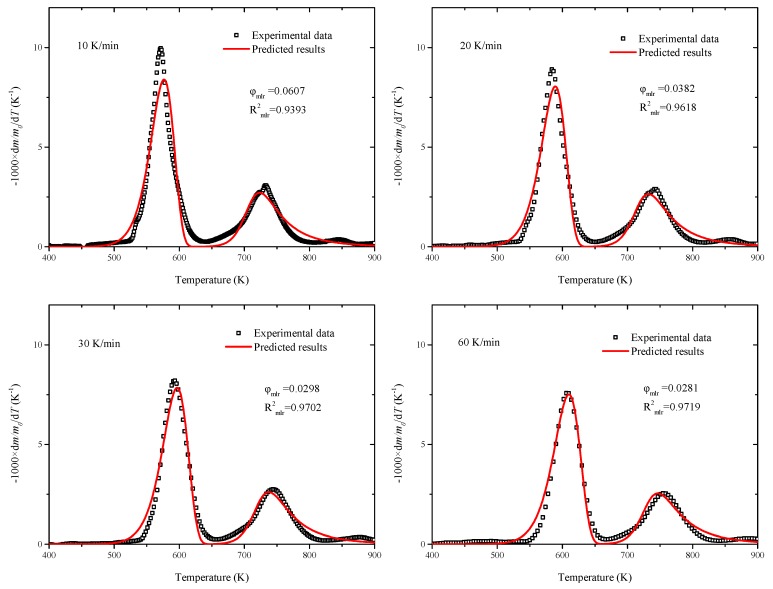
Predicted results based on optimized parameters compared to experimental data.

**Table 1 polymers-11-02080-t001:** Differential and integral expressions of various reaction model functions.

Reaction Model	Differential Form *f(α)*	Integral Form *g(α)*
Power law models		
Power law (P3/2)	*2/3α^−1/2^*	*α^3/2^*
Power law (P2)	*2α^1/2^*	*α^1/2^*
Power law (P3)	*3α^2/3^*	*α^1/3^*
Power law (P4)	*4α^3/4^*	*α^1/4^*
Nucleation models		
Avrami-Erofeev (A2)	*2(1 − α)[−ln(1 − α)]^1/2^*	*[−ln(1 − α)]^1/2^*
Avrami-Erofeev (A3)	*3(1 − α)[−ln(1 − α)]^2/3^*	*[−ln(1 − α)]^1/3^*
Avrami-Erofeev (A4)	*4(1 − α)[−ln(1 − α)]^3/4^*	*[−ln(1 − α)]^1/4^*
Reaction-order models		
First order (F1)	*1 − α*	*−ln(1 − α)*
Second order (F2)	*(1 − α)^2^*	*[1 − (1 − α)^(−1)^]/(−1)*
Third order (F3)	*(1 − α)^3^*	*[1 − (1 − α)^(−2)^]/(−2)*
Fourth order (F4)	*(1 − α)^4^*	*[1 − (1 − α)^(−3)^]/(−3)*
Diffusion models		
1-D diffusion (*D1*)	*(1/2)α^−1^*	*α^2^*
2-D diffusion−Valensi (*D2*)	*[−ln(1 − α)]^−1^*	*(1 − α)ln(1 − α) + α*
3-D diffusion-Jander (*D3*)	*(3/2)[1 − (1 − α)^1/3^]^−1^(1 − α)^2/3^*	*[1 − (1 − α)^1/3^]^2^*
3-D diffusion-Ginstling (*D4*)	*(3/2)[1 − (1 − α)^1/3^]^−1^*	*[1 − (2/3)α]−(1 − α)^2/3^*
Geometrical contraction models		
Prout-Tompkins (*R1*)	*1*	*α*
Contracting cylinder (*R2*)	*2(1 − α)^1/2^*	*1 − (1 − α)^1/2^*
Contracting sphere (*R3*)	*3(1 − α)^2/3^*	*1 − (1 − α)^1/3^*

**Table 2 polymers-11-02080-t002:** Calculation results of *E_a_* and ln*A* by the FWO method.

Stage	α	*E_a_* (kJ/mol)	*R* ^2^	ln*A* [ln(min^−1^)]
**StageI**	0.10	143.41	0.9933	28.44
	0.15	142.47	0.9973	28.33
	0.20	141.19	0.9981	28.11
	0.25	140.14	0.9991	27.93
	0.30	139.29	0.9993	27.77
	0.35	137.81	0.9988	27.47
	0.40	136.05	0.9990	27.08
	0.45	138.70	0.9982	27.57
	0.50	143.39	0.9988	28.42
	Mean value	140.27	0.9980	27.90 *^a^*
**StageII**	0.70	232.68	0.9915	41.77
	0.75	253.13	0.9975	44.94
	0.80	255.24	0.9982	45.27
	0.85	243.24	0.9966	43.48
	Mean value	246.07	0.9959	43.87 *^b^*

*^a^* The value is computed based on the first order reaction-order model. ^*b*^ The value is computed based on the fourth order reaction-order model.

**Table 3 polymers-11-02080-t003:** Calculation results of *E_a_* (kJ/mol) in Stage I based on the CR method.

Reaction Model	10 K/min		20 K/min		30 K/min		60 K/min		Average Value	
	*E_a_*	*R* ^2^	*E_a_*	*R* ^2^	*E_a_*	*R* ^2^	*E_a_*	*R* ^2^	*E_a_*	*R* ^2^
*P3/2*	187.94	0.9572	181.12	0.9671	184.78	0.9606	182.56	0.9755	184.10	0.9651
*P2*	56.33	0.9471	53.92	0.9586	55.02	0.9504	54.61	0.9689	54.86	0.9562
*P3*	34.40	0.9371	32.72	0.9500	33.39	0.9401	32.76	0.9621	33.32	0.9473
*P4*	23.43	0.9241	22.12	0.9386	22.58	0.9264	22.06	0.9528	22.55	0.9355
*A2*	67.93	0.9687	65.07	0.9778	66.45	0.9674	65.35	0.9853	66.20	0.9748
*A3*	42.13	0.9637	40.15	0.9739	41.01	0.9723	40.22	0.9826	40.88	0.9731
*A4*	29.23	0.9575	27.70	0.9690	28.29	0.9612	27.66	0.9791	28.22	0.9667
*F1*	145.33	0.9727	139.81	0.9809	142.76	0.9762	140.73	0.9875	142.16	0.9793
*F2*	171.71	0.9847	165.14	0.9909	168.74	0.9880	166.13	0.9953	167.93	0.9898
*F3*	201.20	0.9920	193.44	0.9963	197.79	0.9951	194.50	0.9989	196.73	0.9956
*F4*	233.61	0.9958	224.54	0.9985	229.71	0.9986	225.66	0.9993	228.38	0.9980
*D1*	253.75	0.9583	244.71	0.9680	249.66	0.9617	246.76	0.9762	248.72	0.9661
*D2*	268.18	0.9641	258.59	0.9731	263.89	0.9675	260.69	0.9807	262.84	0.9713
*D3*	283.97	0.9697	273.75	0.9781	279.44	0.9731	275.91	0.9849	278.27	0.9764
*D4*	273.44	0.9660	263.64	0.9749	269.07	0.9695	265.76	0.9822	267.97	0.9732
*R1*	122.14	0.9550	117.52	0.9652	119.90	0.9584	118.36	0.9741	119.48	0.9632
*R2*	133.34	0.9646	128.28	0.9738	130.94	0.9681	132.94	0.9837	131.37	0.9726
*R3*	137.25	0.9675	132.04	0.9763	134.79	0.9710	129.17	0.9815	133.31	0.9741

**Table 4 polymers-11-02080-t004:** Calculation results of *E_a_* (kJ/mol) in Stage II based on the CR method.

Reaction Model	10 K/min		20 K/min		30 K/min		60 K/min		Average Value	
	*E_a_*	*R* ^2^	*E_a_*	*R* ^2^	*E_a_*	*R* ^2^	*E_a_*	*R* ^2^	*E_a_*	*R* ^2^
*P3/2*	20.98	0.9836	21.42	0.9896	22.46	0.9973	21.55	0.9981	21.60	0.9922
*P2*	−1.01	0.5348	−0.98	0.6524	−0.74	0.7680	−1.17	0.9317	−0.97	0.7217
*P3*	−4.67	0.9880	−4.71	0.9926	−4.60	0.9967	−4.96	0.9980	−4.74	0.9939
*P4*	−6.50	0.9970	−6.58	0.9982	−6.54	0.9990	−6.85	0.9943	−6.62	0.9971
*A2*	13.56	0.9188	13.89	0.9372	14.74	0.9763	13.99	0.9284	14.04	0.9402
*A3*	5.04	0.7727	5.20	0.8229	5.72	0.9352	5.15	0.9284	5.28	0.8648
*A4*	0.78	−0.2026	0.85	−0.0941	1.20	0.4707	0.73	0.1447	0.89	0.0709
*F1*	39.12	0.9587	39.96	0.9680	41.82	0.9875	40.50	0.9879	40.35	0.9755
*F2*	87.28	0.9469	89.15	0.9563	93.09	0.9777	90.70	0.9779	90.05	0.9647
*F3*	150.74	0.9410	153.98	0.9506	160.68	0.9725	156.88	0.9727	155.57	0.9592
*F4*	222.63	0.9394	227.40	0.9488	237.21	0.9708	231.81	0.9710	229.76	0.9575
*F5*	298.04	0.9394	304.42	0.9487	317.47	0.9704	310.40	0.9707	307.58	0.9573
*D1*	31.97	0.9871	32.62	0.9918	34.06	0.9979	32.91	0.9986	32.89	0.9939
*D2*	45.87	0.9822	46.80	0.9877	48.81	0.9971	47.34	0.9976	47.21	0.9911
*D3*	66.66	0.9746	68.03	0.9810	70.91	0.9938	68.98	0.9943	68.65	0.9859
*D4*	52.62	0.9794	53.69	0.9853	55.98	0.9961	54.37	0.9966	54.16	0.9893
*R1*	9.99	0.9694	10.22	0.9807	10.86	0.9950	10.19	0.9963	10.31	0.9854
*R2*	22.25	0.9655	22.73	0.9751	23.88	0.9931	28.23	0.9918	24.27	0.9814
*R3*	27.33	0.9633	27.92	0.9727	29.28	0.9913	22.93	0.9937	26.87	0.9803

**Table 5 polymers-11-02080-t005:** Parameter search range and optimized values by Shuffled Complex Evolution.

Parameters	Calculated Values	Search Range	Optimized Values
*Y_A,0_*	0.50 *^a^*	[0, 1]	0.61
ln*A_A_*[ln (min^−1^)]	27.90	[13.95, 41.85]	29.98
*E_a,A_* (kJ/mol)	140.27	[70.14, 210.41]	146.75
*n_A_*	1.00	-	-
*v_A_*	0.50 *^a^*	[0, 1]	0.35
ln*A_B_*[ln (min^−1^)]	43.87	[21.935, 65.81]	54.94
*E_a,B_* (kJ/mol)	246.07	[123.04, 369.11]	332.81
*n_B_*	4.00	-	-
*v_B_*	0.50 ^a^	[0, 1]	0.44

*^a^* Assumed value.

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
