# Peer review of "Thermal Decomposition Mechanism and Kinetics Study of Plastic Waste Chlorinated Polyvinyl Chloride"

_polymers, 2019, doi:10.3390/polym11122080_

Round 1

Reviewer 1 Report

After analysis of submitted MS “Thermal decomposition mechanism and kinetics study of plastic waste chlorinated polyvinyl chloride”, polymers-657537, please find my observations in order to sustain my decision of revision regarding the publishing of this paper.

The Introduction section: please insert a paragraph regarding the importance of carrying kinetic studies, and present the advantages and disadvantages of model-free and model-dependent methods.

Materials and methods: please insert as section 2.1. Elemental analysis, separate it from Thermogravimetric experiments.

How were the heating rates chosen? Usually, in kinetic studies, the heating rates should be small enough (below 15 K/min) in order to reveal all the thermal events, which aren’t visible at high heating rates due to the thermal inertia of the samples. I would suggest in your future studies to take into account this aspect, since heating rate of 60 K/min are too high for a coherent kinetic analysis!   

Table 1. Please use correctly the superscripts, where necessary. Please take care at details: Avarami should be corrected to Avrami

Results and discussion section: use appropriate superscript and subscript, where necessary, like m0. The data are correctly processed and the kinetic study is believable.

Author Response

Point 1: The Introduction section: please insert a paragraph regarding the importance of carrying kinetic studies, and present the advantages and disadvantages of model-free and model-dependent methods.

Response 1:

Thank you. As you suggested, the following paragraph was added to stress the importance of carrying kinetic studies, and present the advantages and disadvantages of model-free and model-dependent methods in Introduction Section: “Knowledge of pyrolysis kinetics can help provide better understanding and planning of important industrial processes [24], such as their application in pyrolysis model [25] and direct combustion [26]. Model-fitting and model-fitting methods are the common ways to explore the kinetic parameters. Model-fitting method consists of fitting different models to the experimental data for the best statistical fit but with the inability to determine the reaction model [27]. However, model-free method can come over this problem without prior knowledge of the reaction model, and estimate the kinetic parameters at specific extent of conversion to provide appropriate search ranges for model-fitting method [28]. Then it is recommended to explore the kinetic parameters by coupling both the model-fitting and model-free method.”

Point 2: Materials and methods: please insert as section 2.1. Element alanalysis, separate it from Thermogravimetric experiments.

Response 2:

Thank you for the suggestion. We inserted the Elemental analysis as Section 2.1 and separated it from Thermogravimetric experiments. Please see the attachment.

Point 3: How were the heating rates chosen? Usually, in kinetic studies,the heating rates should be small enough (below 15 K/min) inorder to reveal all the thermal events, which aren’t visible at high heating rates due to the thermal inertia of the samples. I would suggest in your future studies to take into account this aspect, since heating rate of 60 K/min are too high for a coherent kinetic analysis!

Response 3:

Thank you for the suggestion. We will take the lower heating rates into account for kinetic analysis in our future studies.

Point 4: Table 1. Please use correctly the superscripts, where necessary. Please take care at details: Avarami should be corrected to Avrami.

Response 4:

Thank you for the suggestion. We had adjusted the superscripts and corrected Avarami to Avrami in Table 1. Please see the attachment.

Point 5: Results and discussion section: use appropriate superscript andsubscript, where necessary, like m0. The data are correctly processed and the kinetic study is believable.

Response 5:

Thank you for the suggestion. We had corrected the superscript and subscript in Results and discussion section. Please see the attachment.

Reviewer 2 Report

The article entitled “Thermal decomposition mechanism and kinetics 2 study of plastic waste chlorinated polyvinyl 3 chloride” by Ru Zhou and co-workers concern the study of thermal decomposition of CPVC which was widely studies in the literature already. Simply shifting the angle from fire retardant properties of the polymer to the problem of engineering plastic waste, does not create scientific novelty. The novelty of this research is in application of kinetic modeling to CPVC thermal decomposition processes. However, the written objective of the study is rather obscure and lack proper emphasis of the real aim of the study. The results of modeling are not discussed in the light of known facts about the degradation mechanism from the literature.

In general the work is interesting but I suggest to improve significantly the discussion of result in the context with published mechanisms of the processes. Kinetics of the stage II which is reported to involve simultaneous crosslinking and intramolecular decomposition of the polyene segments, can  occur via polyene free radicals or via Diels -Alder cyclisation process. I would expect the discussion to this point and explanation which of the hypothetical mechanisms can be supported based on the modeling. And there are more point that can improved in the discussion section.

The conclusions part is rather a summary of facts, while it should address the aim of the study and add some value to the knowledge about the reactions.

Summing up I suggest to return the manuscript to the authors for substantial revision

Author Response

Point 1: The article entitled “Thermal decomposition mechanism and kinetics study of plastic waste chlorinated polyvinyl chloride” by Ru Zhou and co-workers concern the study of thermal decomposition of CPVC which was widely studies in the literature already. Simply shifting the angle from fire retardant properties of the polymer to the problem of engineering plastic waste, does not create scientific novelty. The novelty of this research is in application of kinetic modeling to CPVC thermal decomposition processes. However, the written objective of the study is rather obscure and lack proper emphasis of the real aim of the study. The results of modeling are not discussed in the light of known facts about the degradation mechanism from the literature.

In general the work is interesting but I suggest to improve significantly the discussion of result in the context with published mechanisms of the processes. Kinetics of the stage II which is reported to involve simultaneous crosslinking and intramolecular decomposition of the polyene segments, can occur via polyene free radicals or via Diels -Alder cyclisation process. I would expect the discussion to this point and explanation which of the hypothetical mechanisms can be supported based on the modeling. And there are more point that can improved in the discussion section.

Response 1:Thank you for the suggestion. Just as you said, the main novelty of this research is in the application of kinetic modeling to CPVC thermal decomposition processes. The detailed and deeper kinetic mechanism, such as the crosslinking and intramolecular decomposition in Stage II, should be further explored to establish its process (polyene free radicals or Diels -Alder cyclisation). However, it is hard to speculate or validate the deeper kinetic mechanism, just by our current thermogravimetric experimental data. Then the mechanism of the reaction was not discussed in detail and the main aim of our research is to establish the reaction kinetic parameters, which can be used to provide better understanding and planning of important energy utilization or industrial processes, especially their application in pyrolysis model and direct combustion in our previous study. So the following sentences were added to Introduction: “However, the above researches are not enough to reveal the pyrolysis dynamics of engineering plastic waste CPVC, so the aim of the current paper is to explore its pyrolysis behaviors and obtained appropriate kinetic parameters. Knowledge of pyrolysis kinetics can help provide better understanding and planning of important industrial processes [24], such as their application in pyrolysis model [25] and direct combustion [26].” and Section 3.4: “The optimized kinetic parameters can be used in the pyrolysis process of related energy utilization, such as the application of pyrolysis model and direct combustion in numerical simulation [25, 26].”

Point 2: The conclusions part is rather a summary of facts, while it should address the aim of the study and add some value to the knowledge about the reactions.

Response 2:

Thank you for the suggestion. Furthermore, the Conclusion Section was revised and the aim of this study was stressed. Then the following sentences were added to the conclusion section: “Then these obtained kinetic parameters can be used in the further simulation about pyrolysis and combustion.”

Round 2

Reviewer 2 Report

I'm satisfied with the authors' answers and introduced corrections. I recommend to accept the manuscript